# Incidence of Aflatoxin M$_1$ in Milk and Milk Products from Punjab, Pakistan, and Estimation of Dietary Intake

**Shahzad Zafar Iqbal \*** **, Muhammad Waqas**  **and Sidra Latif**

Department of Applied Chemistry, Government College University, Faisalabad 38000, Pakistan
* Correspondence: shahzad@gcuf.edu.pk; Tel.: +92-300-7684577

**Abstract:** In the present study, 124 samples of milk and milk products were analyzed for the presence of aflatoxin M$_1$ (AFM$_1$), which were purchased from the central cities of Punjab, Pakistan. The analysis was carried out using reverse-phase liquid chromatography, which was equipped with a fluorescence detector. The results showed that 66 samples (53.8%) of raw milk and milk products were found to be contaminated with detectable levels of AFM$_1$ above $\leq$50 ng/L, and 24.2% of the samples had levels of AFM$_1$ higher than the permissible limit of the European Union (EU; 50 ng/kg). In total, 53.6% of the raw milk, 57.8% of the UHT (ultra-heat-temperature) milk, 45% of the powdered milk, 57.1% of the yogurt, 55.5% of the cheese, and 50% of the buttermilk samples had levels higher than the LOD, i.e., 4 ng/L. The highest mean of 82.4 $\pm$ 7.8 ng/kg of AFM$_1$ was present in the positive samples of raw milk. The highest dietary intake of AFM$_1$ was found in infants' milk (5.35 ng/kg/day), UHT milk (1.80 ng/kg/day), powdered milk (5.25 ng/kg/day), and yogurt (1.11 ng/kg/day). However, no dietary intake was detected in the cheese and butter milk samples used for infants. The results from the undertaken work are beneficial for establishing rigorous limits for AFB$_1$ in animal feed, especially considering the high prevalence rate of hepatitis cases in the central cities of Punjab, Pakistan.

**Keywords:** milk; milk products; aflatoxin M$_1$; dietary intake; HPLC



## 1. Introduction

Milk is known as a highly nutritious and balanced food, especially for children [1]. According to the FAO report, global milk production was forecast at 852 million tons in 2019, an increase of 1.4% from the previous year, and significant contributors are India, Pakistan, China, and the European Union countries [2]. Pakistan is ranked 4th in milk production with 58 billion liters per annum. Considering Pakistan is an agricultural country, its population of 55 million are directly dependent on livestock and agriculture. The most abundant milk production is produced by buffalos [3,4]. Factors, such as advancements in the dairy industry, resources and training, good storage, and preservation practices of dairy products, could bring a revolution in the dairy industry. Furthermore, the most demanding task is to maintain the quality and safety of milk products for consumers [5].

Aflatoxins are the most important class of mycotoxins, produced by the fungi *Aspergillus flavus*, *Aspergillus parasiticus*, and *Aspergillus nominus* [6]. They can contaminate various food products during the pre-harvest and postharvest stages. Their presence in food has resulted in global concern over the quality and safety of food due to their toxicity and carcinogenicity [2,7–9]. Some studies have documented that exposure to AFs in humans has synergistic effects on consumers with liver carcinogens [10], and hepatocellular carcinoma (HCC) is one of the highest causes of deaths with cancer [11]. In Pakistan, 10 million people are affected by hepatitis C (5% of the total population), the second highest in the world, and the prevalence rate of hepatitis C patients in Faisalabad is 25.1% [12,13]. Aflatoxin B$_1$ metabolites are present in the form of AFM$_1$ in the body, which is mostly secreted in the milk and urine of animals [14]. There have been reports that show a linear relationship between the content of AFM$_1$ in milk and AFB$_1$ consumption through

feedstuffs. It has been observed that 0.5 to 6% of $AFB_1$ is converted into $AFM_1$ during metabolite formation. After injection of $AFB_1$ with contaminated food in lactating animals, $AFM_1$ appears 12 to 24 h after in milk, and reaches the maximum levels excreted in milk samples after 72 h [6,14]. The International Agency for Research on Cancer (IARC) classifies $AFM_1$ as a group 2b a carcinogenic agent, and the toxicity of $AFM_1$ is 10% less than its parent compound, i.e., $AFB_1$ [15]. Considering the toxicity and carcinogenic nature of this toxin, various regulatory agencies have implemented a permissible limit for $AFM_1$. The US Food and Drug Administration has established the recommended legal limit of 0.5 ng/L in milk [16]. The European Union has established a 50 and 25 ng/L permissible limit for $AFM_1$ in milk intended for adults and infants, respectively [17]. Although, no harmonized limit has been established or implemented for $AFM_1$ in milk and milk products [18].

In our preceding reports [4,6,19–21], a substantial amount of $AFM_1$ has been observed in milk and its products. Faisalabad is the third central populous city of Pakistan. High amounts of $AFM_1$ in milk and its products might cause serious health hazards to local populations, considering the city has the highest rate of hepatitis. The present research aimed (i) to investigate the occurrence of $AFM_1$ in milk and milk product (cheese, buttermilk, UHT milk, and powdered milk) samples from two main milk-producing areas, i.e., Faisalabad and Jhang, from Punjab, Pakistan, and to compare the levels of $AFM_1$ with the permissible limits of the EU; and (ii) to investigate the dietary intake of $AFM_1$ in milk and milk products in different age groups of consumers. The findings of the undertaken survey will assist food regulatory agencies to take necessary measures and implement strict regulations.

## 2. Materials and Methods

### 2.1. Sampling

A total of 124 samples of milk and its products (milk (24), UHT milk (19), powdered milk (20), yogurt (21), cheese (18) and buttermilk (18)) were collected from dairy farms, milk shops, and supermarkets in Faisalabad and Jhang cities of Punjab, Pakistan, during April 2019. The milk and milk products mostly consisted of buffalo milk. The UHT milk samples were purchased from superstores, shops, and markets. These regions are the main milk-producing cities in Punjab, Pakistan. The liquid samples were at least 1 L, and the solid samples were 1 kg each. The samples were collected in plastic bags and were kept in the freezer at −20 °C until further analysis.

### 2.2. Chemicals and Reagents

The solvents, i.e., acetonitrile (99%), methanol (99%), and $AFM_1$ (10 mg/L, 2 mL in acetonitrile, 99.9%), were purchased from Sigma-Aldrich (Baden-Württemberg, Germany). The columns used for $AFM_1$ extraction from milk and milk products were immuno-affinity (Afla $M_1$ FL+; VICAM, Watertown, NY, USA). The reagents and chemicals used in this study were freshly prepared, and the purity of all chemicals was above 85%. Double-distilled water was used throughout this study.

### 2.3. Extraction of Aflatoxin $M_1$

The process of sample extraction of $AFM_1$ from raw milk was carried out using our earlier method (20). Briefly, the milk samples were heated at 37 °C (in a water bath) and then centrifuged at 3500 rpm for 5 min to separate the fat layer. Then, the samples were filtered using Whatman No. 5 filter paper, and 50 mL filtrate was transferred to the syringe barrel of IAC and passed at a rate of 2 mL/min using a solid-phase extraction manifold. The column was washed with 20 mL double-distilled water, and $AFM_1$ was eluted with 4 mL pure acetonitrile, passing the IAC in approximately 60 s. Finally, the eluate was evaporated to dryness using a gentle stream of nitrogen at 40 °C. The residue was dissolved in 1 mL of the mobile phase, and 20 μL solution was injected into HPLC analysis. The extraction was carried out in yogurt and cheese samples as discussed by Iqbal and Asi [4]. Briefly, 10 g sample (cheese, powdered milk, yogurt, and buttermilk) and 10 g Celite (Sigma-Aldrich,

St. Louis, MO, USA) were added to 80 mL of dichloromethane and blended for 3 min. Then, the mixture was centrifuged at 21,000 rpm for 4 min to form a slurry. After centrifugation, the slurry was filtered with Whatman no. 5 filter paper, and the filtrate was evaporated to dryness under nitrogen stream at 40 °C. Then, the same procedure used for the extraction of $AFM_1$ from milk or UHT milk samples was followed.

### 2.4. HPLC Conditions

System liquid chromatography (Shimadzu, series LC-10A, Kyoto, Japan) and a fluorescence detector (model RF-530) were used. The excitation (365 nm) and emission (435 nm) wavelengths were set before the experiment. The Discovery (Supelco, Bellefonte, PA, USA) C18 (4.68 × 250 mm, 5 mm) column was used during the analysis. The 25% mixture of acetonitrile with 75% of water was used as the mobile phase with a flow rate of 1.3 mL/min.

### 2.5. Dietary Intake Estimation

The detection of dietary intake of $AFM_1$ in milk and milk products was carried out following our previous method [20]. The dietary intake questionnaires about the milk and milk product consumption of infant, male, and female individuals were distributed randomly among 800 participants. Their responses about the consumption of milk and milk-related products were evaluated and the mean consumption of milk and milk products during the last 4 weeks was evaluated. Accurate collection information from the interviewees, including the exact size of the bowl, glass, and packs, was noted, and shown to arouse the interviewees' memory. However, eating habits, seasons, and cultural difference might cause variation in the results of the dietary intake estimation.

To assess the $AFM_1$ content of dairy products, the average level in each dairy product was calculated by taking in account both positive and negative results and using LOD/2 for samples with levels lower than LOD [1]:

$$\text{Dietary intake ng/Kg/day} = \frac{\text{Consumption of milk \& product}\left(\frac{\text{L}}{\text{day}}\right) \times \text{Levels of AFM1 in milk and product}\left(\frac{\text{ng}}{\text{L}}\right)}{average\ individual\ weight(kg)}$$

### 2.6. Statistical Analysis

The samples were analyzed as triplicate, and the results are presented as mean ± SD. Regression/correlation analysis was used to determine the value of $R^2$ (the equation of the straight line was constructed using Excel MS 365), and significant differences in the $AFM_1$ levels in milk and its products were determined using one-way ANOVA ($\alpha = 0.05$) and LSD was used to evaluate significant differences between each treatment (SPSS Statistics 19 software, Chicago, IL, USA).

## 3. Results

### 3.1. Method Validation

The functional relationship between the response to the instrument and the concentration of $AFM_1$ was analyzed by constructing a six-point standard curve, i.e., 10, 20, 40, 80, 100, and 200 ng/L. The coefficient of determination was $R^2$ 0.9985 and the LOD and LOQ were 4 and 8 ng/L, respectively. The recovery of the method was assessed by adding 25, 50, and 100 ng/L of $AFM_1$ to the milk and milk samples. The results showed good recoveries, i.e., 70 to 91% with a relative standard deviation (RSD) that varied from 10 to 25%, as presented in Table 1.

**Table 1.** Recovery percentage of aflatoxin $M_1$ in milk and milk products from Punjab, Pakistan.

| Added Level of $AFM_1$ ng/L | Milk | | UHT Milk | | Powdered Milk | | Cheese | | Buttermilk | |
|---|---|---|---|---|---|---|---|---|---|---|
| | Mean [a] ng/L | RSD % | Mean [a] ng/L | RSD % | Mean [a] ng/kg | RSD % | Mean [a] ng/kg | RSD % | Mean [a] ng/L | RSD % |
| 25 | 19 (76) | 13 | 18 (72) | 17 | 20 (80) | 19 | 16 (64) | 25 | 18 (72) | 16 |
| 50 | 041 (82) | 15 | 39 (78) | 14 | 35 (70) | 24 | 36 (72) | 17 | 38 (76) | 14 |
| 100 | 90 (90) | 10 | 89 (89) | 21 | 90 (90) | 10 | 88 (88) | 13 | 91 (91) | 10 |

[a] Mean of 4 replicates of each spiked concentration; parentheses represent the percent recovery.

### 3.2. Occurrence of $AFM_1$ in Milk and Milk Products

The occurrence of $AFM_1$ was examined in 124 samples of raw milk, and milk products (as mentioned in the sampling section) were collected from Punjab, Pakistan (Table 2). Samples were considered positive when they had an average amount of $AFM_1$ above the LOD ($\geq$4 ng/kg). The findings demonstrate that out of 124 samples, 66 (53.2%) samples were found to be contaminated with $AFM_1$, and the amount ranged from LOD to 210 ng/L. The highest mean contamination was found in raw milk samples, i.e., 82.4 $\pm$ 7.8 ng/L, and the lowest amount was documented in buttermilk, i.e., 41.5 $\pm$ 5.4 ng/L. The frequency of the samples with $AFM_1$ levels higher than the EU permissible limits is shown in Table 3. In total, 24.2% of the samples had levels higher than 0.50 ng/g (the EU-recommended limits for milk). Furthermore, 16.1% of the samples had $AFM_1$ levels greater than 100 ng/L.

**Table 2.** Incidence and contamination level of aflatoxin $M_1$ in milk and milk products from Punjab, Pakistan.

| Types | Total Samples *n* | Positive *n* (%) | Mean (ng/L) $\pm$ S.D. | Range (µg/L) |
|---|---|---|---|---|
| Milk | 28 | 18 (64.2) | 82.4 $\pm$ 7.8 | <LOD-210 |
| UHT milk | 19 | 11 (57.9) | 68.7 $\pm$ 6.5 | <LOD-180 |
| Powdered milk | 20 | 9 (45.0) | 60.5 $\pm$ 7.1 | <LOD-150 |
| Yogurt | 21 | 12 (57.1) | 55.8 $\pm$ 9.6 | <LOD-110.5 |
| Cheese | 18 | 10 (55.6) | 52.7 $\pm$ 8.4 | <LOD-98.7 |
| Buttermilk | 18 | 9 (50.0) | 41.5 $\pm$ 5.4 | <LOD-90.5 |
| Total | 124 | 69 (55.6) | | |

Limit of detection = LOD; LOQ = limit of quantification. LOD = 4 ng/kg; LOQ = 8 ng/kg. Mean concentration ng/L for milk and ng/kg for cheese and yogurt samples.

**Table 3.** Frequency of milk and milk product samples that exceeded EU limits.

| Types | $n \leq 50$ ng/L | $n$ (50–100 ng/L) | $n \geq 100$ ng/L | $n \geq$ EU Limits [a] |
|---|---|---|---|---|
| Milk | 8 | 7 (21.0) | 3 | 7 (25.0) |
| UHT milk | 4 | 7 (37.0) | 2 | 7 (36.8) |
| Powdered milk | 5 | 4 (20.0) | 3 | 4 (20.0) |
| Yogurt | 7 | 5 (29.0) | 3 | 5 (23.8) |
| Cheese | 4 | 6 (33.0) | 5 | 2 (11.1) |
| Buttermilk | 4 | 5 (28.0) | 4 | 5 (27.7) |
| Total | 32 (25.8) | 34 (27.4) | 20 (16.1) | 30 (24.2) |

[a] The EU limit is 50 ng/L or ng/kg. All samples are within the legal limit of USDA, i.e., 500 ng/L. Parentheses (% of samples).

Furthermore, the amounts of $AFM_1$ in raw milk and milk products from two cities, i.e., Faisalabad and Jhang, are shown in Table 4. The amounts of $AFM_1$ in milk and cheese samples from Faisalabad and Jhang cities were found to be significantly different (at $\alpha$ = 0.05) while the levels in other milk products were not statistically significant. Furthermore, 32.2% of the samples of milk and milk products from Faisalabad city and 24.2% of the samples from Jhang city were higher than the permissible EU limits.

**Table 4.** Incidence and contamination level of aflatoxin $M_1$ in milk and milk products from the Faisalabad and Jhang regions.

| | Faisalabad | | | | | Jhang | | | | |
|---|---|---|---|---|---|---|---|---|---|---|
| **Types** | **Samples** $n$ | **Positive** $n$ **(%)** | **Mean (ng/L)** $\pm$ **S.D.** | **Range (µg/L)** | $n \geq$ **EU (%)** | **Sample** $n$ | **Positive** $n$ **(%)** | **Mean (ng/L)** $\pm$ **S.D.** | **Range (µg/L)** | $n \geq$ **EU (%)** |
| Milk | 16 | 10 (62.5) | 89.2 ± 6.3 [a] | LOD- 198 | 5 (31.2) | 12 | 8 (66.6) | 76.5 ± 8.4 [b] | <LOD- 195 | 3 (25.0) |
| UHT milk | 10 | 5 (50.0) | 70.5 ± 4.3 [a] | LOD- 210 | 4 (40.0) | 9 | 6 (66.7) | 64.4 ± 7.6 [b] | <LOD- 209 | 2 (22.2) |
| Powdered milk | 9 | 5 (55.6) | 63.5 ± 9.4 [a] | LOD- 180 | 1 (11.1) | 11 | 4 (36.4) | 57.3 ± 10.1 [b] | <LOD- 198 | 0 |
| Yogurt | 11 | 7 (63.6) | 61.2 ± 6.5 [a] | LOD- 120.3 | 2 (25.0) | 10 | 5 (50.0) | 52.5 ± 6.9 [b] | <LOD- 115.4 | 3 (23.0) |
| Cheese | 10 | 6 (60.0) | 58.5 ± 6.7 [a] | LOD- 110.6 | 4 (40.0) | 8 | 4 (50.0) | 43.4 ± 6.5 [b] | <LOD- 163.3 | 4 (50.0) |
| Buttermilk | 9 | 5 (55.6) | 49.3 ± 7.3 [a,*] | LOD- 98.5 | 4 (44.4) | 9 | 4 (44.4) | 39.7 ± 7.8 [b,*] | <LOD- 99.5 | 3 (33.3) |
| Total | 62 | 38 (61.2) | | | 20 (32.2) | 62 | 31 (50.0) | | | 15 (24.2) |

$n$ = number of samples. LOD = limit of detection (4 ng/L); LOQ = 8 ng/L. S.D. = standard deviation. [a,b] The means of data within columns with different English alphabetic letters show significant differences in $AFM_1$ among the different types of milk products ($\alpha$ = 0.05). [a,b] * The means of data within columns with different English alphabetic letters show significant differences in $AFM_1$ among the different types of milk products ($\alpha$ = 0.01).

## 4. Discussion

### 4.1. Occurrence of AFM₁ in Milk, UHT Milk, and Powdered Milk

Higher levels of $AFM_1$ in raw milk (212.2 ng/L) were documented from Pakistan [4]. In another, study the levels of $AFM_1$ in raw milk, UHT milk, and powdered milk were 94.9, 75.2, and 65.1 ng/L, respectively, in Pakistan [20]. However, comparatively lower levels of $AFM_1$ in raw milk (73 ng/L) and UHT milk (60 ng/L) were observed in Pakistan [21]. In the neighboring country Iran, the mean levels of $AFM_1$ in pasteurized and UHT milk samples were 52.5 and 46. 4 ng/L, respectively, with the concentrations ranging from 5.8 to 528.5 and 5.6 to 515.9 ng/L, respectively. In this study, 31 and 19 samples of pasteurized milk and UHT milk had levels of $AFM_1$ higher than the EU legal limit [22].

Studies from other countries [23–25] have observed elevated amounts of $AFM_1$ in raw milk and UHT milk samples. From Sudan, a high mean level of 2070 ng/L of $AFM_1$ in raw milk samples was observed, with the concentration ranging from 220 to 6900 ng/L [25]. Daou et al. [23], from Lebanon, observed that 58.8% of the milk samples had amounts higher than the LOD (mean 113 ng/L, levels ranged from 11 to 440 ng/L) and 28% of the samples had concentrations higher than the EU-recommended limits. However, a very high percentage (90.9%) of UHT milk samples were contaminated with $AFM_1$, with an average of 69 ng/L, and the concentration ranged from 13 to 219 ng/L. Bahrami et al. [24], from Iran, documented that 68.5% of raw cow milk samples had levels of $AFM_1$ higher than LOD, with levels ranging from 19 to 203.4 ng/L.

In other studies, lower amounts of $AFM_1$ in milk and UHT milk samples were observed compared to the findings of the present research [6,19,21,26–36]. In Pakistan, a level of 10 to 200 ng/L in raw milk was documented [6]. An LOD to 510 ng/L in raw and UHT milk samples [21], a mean level of 55 ng/L in raw milk [19], a meaningful amount of 37.4 ng/L in milk [26], and a mean level of 16.01 ng/L in milk and UHT milk samples from Qatar [27] have been reported.

Studies from Pakistan, China, Sudan, Qatar, Turkey, and Iran have demonstrated high levels of $AFM_1$ in milk samples. The geographical and environmental conditions of these countries have include drought periods, high moisture levels, and relatively high temperatures during the summer, and very low temperature during winter seasons were recorded. These factors provide favorable conditions for the growth of the fungi *Aspergillus* to contaminate animal feed and produce aflatoxin $B_1$ [36]. Furthermore, other factors such as the adoption of good agricultural practices, variation in animal grazing systems, transportation management, farm management practices, storage, and good analytical analysis might be effective in reducing the levels of $AFM_1$ in milk samples [6]. Recently, the Punjab Food Authority was established to control and maintain the quality of food in Punjab, Pakistan.

### 4.2. Occurrence of AFM₁ in Yogurt, Cheese, and Buttermilk Samples

The levels of $AFM_1$ in yogurt (55.8 ± 9.6 ng/kg) can be compared to the findings of our earlier reports, i.e., 63.6 and 59.6 ng/kg in milk samples from the winter and summer seasons, respectively [20]. Furthermore, higher concentrations, compared to the present results, were found in yogurt samples from Pakistan: 90.4 ± 11.1 ng/kg [21]. In another study, comparable levels of $AFM_1$ in yogurt samples (53 ng/L) and butter samples (36 ng/L) were reported [21]. In China, 15 samples of yogurt were found to be positive for $AFM_1$, with concentrations of 17.2 ± 9.5 ng/kg [26]. In Malaysia, five samples of yogurt were used for the analysis of $AFM_1$. Only two samples were found to be positive for $AFM_1$, with concentrations of 25.5 ± 7.2 ng/kg [28]. Higher amounts of $AFM_1$ in cheese samples as compared to milk samples were reported in previous studies [23,24,26,27,33]. In Iran, a lower level of $AFM_1$ in cheese samples as compared to milk samples was reported [24]. The variation in the levels of $AFM_1$ in cheese compared to milk might be due to the processing steps involved during the production of cheese. A slightly low pH and the presence of lactic acid bacteria in the yogurt sample might be responsible for the variation in $AFM_1$ [37].

Furthermore, factors such as the initial amount of $AFM_1$ in milk; the differences in the extraction techniques, yogurt type, storage temperatures, and durations; and the cultures used for yogurt production are crucial for the variation in $AFM_1$ [28]. Glucose, during storage, is oxidized by the enzyme glucose oxidase, which produces hydrogen peroxide and gluconolactone. The hydrolysis of gluconolactone produces gluconic acid (pH 3.9) and the low pH might be responsible for the degradation of $AFM_1$ in yogurt samples [38].

It is evident from the results that significant differences exist in the levels of $AFM_1$ in dairy samples from Faisalabad compared to the samples from Jhang (Table 4). This variation in the levels of $AFM_1$ might be explained by the fact that Faisalabad is an industrial city as compared to Jhang. Therefore, local farmers use more concentrated feed than green fodder [6]. A low amount of $AFM_1$ in the buttermilk samples, i.e., 39.7 $\pm$ 7.8 ng/kg, from Jhang was observed in the present study compared to the levels of $AFM_1$ in butter samples (69.7 ng/kg) in a previous study [21].

### 4.3. Dietary Intake Estimation

The intake of $AFM_1$ from milk and milk products of different age groups, i.e., infant, female, and male individuals, is documented in Table 5. The highest contributor to the dietary intake of $AFM_1$ was found to be raw milk, representing 40%, 27%, and 30% of the intake of infants, females, and males, respectively. The average dietary intake of $AFM_1$ in infants, females, and males was found to be 5.35, 2.73, and 2.94 ng/kg/day, respectively. However, a higher dietary intake value of 6 ng/kg/day for infants' milk was documented in Servia [39]. However, lower levels of dietary intake compared to the findings of the presented study were documented, i.e., 0.054 ng/L/day in raw milk [18,40].

**Table 5.** Dietary intake of aflatoxin $M_1$ level in different seasons and in different age groups.

| | Milk | | | UHT Milk | | | Powdered Milk | | | Yogurt | | | Cheese | | | Buttermilk | | |
|---|---|---|---|---|---|---|---|---|---|---|---|---|---|---|---|---|---|---|
| | Intake L/Day | Lowest ng/L/Day | Highest ng/L/Day | Intake L/Day | Lowest ng/L/Day | Highest ng/L/Day | Intake L/Day | Lowest ng/L/Day | Highest ng/L/Day | Intake L/Day | Lowest ng/L/Day | Highest ng/L/Day | Intake L/Day | Lowest ng/L/Day | Highest ng/L/Day | Intake L/Day | Lowest ng/L/Day | Highest ng/L/Day |
| Infants | 0.51 | 2.10 | 5.35 | 0.20 | 0.68 | 1.80 | 0.70 | 2.12 | 5.25 | 0.20 | 0.55 | 1.11 | 0 | 0 | 0 | 0 | 0 | 0 |
| Males | 0.91 | 1.07 | 2.73 | 0.60 | 0.58 | 1.54 | 0.40 | 0.34 | 0.86 | 0.90 | 0.72 | 1.42 | 0.50 | 0.38 | 0.71 | 1.50 | 0.88 | 1.94 |
| Females | 0.70 | 1.15 | 2.94 | 0.50 | 0.68 | 1.80 | 0.30 | 0.36 | 0.90 | 0.50 | 9.55 | 1.10 | 0.35 | 0.36 | 0.69 | 0.90 | 0.74 | 1.62 |

Lowest dietary intake = mean level of $AFM_1$. Highest dietary intake = highest level of $AFM_1$. Infant average weight = 20 (kg), male average weight = 70 (kg), female average weight = 50 (kg).

## 5. Conclusions

In the current survey, the amounts of AFM$_1$ in milk and milk products were found to be relatively lower than the findings of our previous studies. The results showed that 55.6% of the samples of milk and milk products were found to be contaminated with AFM$_1$, with levels ranging from the LOD to 90.5 and the LOD to 210 µg/kg. The dietary intake of AFM$_1$ for infants was found to be relatively high at 5.35 ng/kg/day in raw milk used for infants. Following suggestion are recommended

i.   It is recommended that continuous monitoring should be initiated for the detection of AFB$_1$ in animal feed and fodder and for AFM$_1$ in milk and milk products.
ii.  The relatively high levels of AFM$_1$ in dairy food should be communicated with stakeholders, including farmers, trader, milkmen, and people working in the dairy industry.
iii. The regulatory agencies should implement strict regulations for these toxins.

**Author Contributions:** S.Z.I., conceptualization, funding acquisition, supervision, writing, and editing, S.L., validation and formal analysis; M.W. validation and visualization. All authors have read and agreed to the published version of the manuscript.

**Funding:** The study appreciate the funding provided by HEC [Grant no. NRPU-5574]. The APC was waived off by Journal.

**Institutional Review Board Statement:** The study needs no certificate from institutional board.

**Informed Consent Statement:** The consent has been taken from all participants during data collection of food frequency questionnaire.

**Data Availability Statement:** The data will be available when requested.

**Acknowledgments:** The authors thank HEC, Islamabad, for providing funds to conduct the present research and analytical tasks supported by NIAB, Faisalabad, Pakistan.

**Conflicts of Interest:** The authors declared no conflict of interest.

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
