# Peer review of "Incidence of Aflatoxin M1 in Milk and Milk Products from Punjab, Pakistan, and Estimation of Dietary Intake"

_2624-862X, doi:10.3390/dairy3030041_

Round 1
Reviewer 1 Report
Dear authors, the work is of relevance and interest to the local population. However, the work needs to be improved in its writing form. The writing doesn't sound scientific and the English needs a thorough review. The research objective must be presented at the end of the introduction. Review UTH by UHT throughout the text. Combine the results with each product's respective discussion to improve the reading experience. Table 5 does not make sense to be presented, they are not your results, and this is not a review article.
Author Response
The file has been uploaded

Reviewer 2 Report
The manuscript by Iqbal et al provides information on the incidence of aflatoxin M1 in milk and milk products from Punjab, Pakistan. The justification for this research should be clearly stated especially because similar studies of which some of the authors of the present study were involved was previously published as stated by the authors (introduction: line 79 - 79). I sometimes found it difficult to follow through some aspects of the manuscript. For example, the entire abstract should be re-written for clarity. In addition, the exposure assessment section should be carefully looked into. For example, what is the dietary intake of milk consumers in the present study. How was the exposure assessment determined for males and females as stated in the line 78-79, page 14.
Below are some specific comments:
Line 27: please add "in" before total. Delete "one hundred and twenty-four", the number "124" should be sufficient. "Arranged" or purchased. Kindly clarify if there are more than one central cities in Punjab.
Line 28: examined for the "detection" or "presence of"
Line 29: again add "in" before total and delete "sixty-six"
Line 30: "its diary products" suggest the products analyzed were obtained directly from the raw milk analysed.
Line 32: kindly provide the EU limit
Line 32: please define UHT at first mention.
Line 44: "shown"?
Line 46: samples "24.2%"?
Line 57: please provide a reference
Line 59-61: please rewrite for clarity
Line 66: Afs are "associated with liver carcinogen"?
Line 71: AFM1 is a class I carcinogen?
Line 126-134: please rewrite for clarity and cite the appropriate references. For example questionnaire were distributed to 800 participants, how are these respondents related to the present research? What is the average weight of diary product consumed? Perhaps, this was used in calculating the daily intake. Of course the weight of food consumed by infants will be different from adults. All these information should be provided.
Line 168: LOD or <LOD, please correct throughout the manuscript.
Table 5: please replace "to" by "-"
Line 25, Page 12: what do authors mean by "elevated levels"
Line 78-79: how was daily intake for females and males calculated?
Reviewer 3 Report
Dear editor
This manuscript entitled "Incidence of aflatoxin M1 in milk and milk products from Punjab, Pakistan and estimation of dietary intake" was discussed a good idea
However, 1- introduction need to enrich by more related reference to the issue.
2 - Result need to present more clearly way.
3- The section of discussion need to be more focused.
4- Conclusion need to be shorten and direct to the recommendation.
Reviewer 4 Report
Review
for the journal Dairy (ISSN 2624-862X)
Manuscript ID: dairy-1747999
Title: Incidence of aflatoxin M1 in milk and milk products from Punjab, Pakistan and estimation of dietary intake
Authors: Wirot Likittrakulwong, Pisit Poolprasert, Worawatt Hanthongkul and Sittiruk Roytrakul
1) In general, the topic of the article is quite interesting, as it is devoted to the study of the occurrence of AFM1 in milk and its products. However, some concerns about the statistical approach and the construction of discussion and conclusion chapters reduce the scientific quality of the paper. In addition, the objectives of the study are not clearly formulated and their relationship is not sufficiently covered.
2) Lines 27-29. “Total 124 One hundred and twenty-four samples”. “Total 66Sixty-six” Choose numbers or text.
3) Line 30. “were having levels had detectable levels of 30 AFM1 (above ≥ 4 ng/L)”. Parentheses are not needed.
4) The text of the summary is inconsistent, difficult to understand and unreasonable conclusion about the usefulness of the results. Authors are encouraged to submit the article to an English language specialist.
5) Lines 85-86. “The findings of the undertaken survey will assist peoples related to the dairy industry to take necessary measures”. It is unclear how this will be achieved. Did you make recommendations to specific institutions? Need clarification.
6) 2.1. Sampling. It is not clear if all dairy products were from buffalo milk.
7) Lines 127-134. “The detection of dietary intake of AFM1 in milk and milk products was carried out following our previous method”. This is an important part of your research and should be commented on in more detail. The sample design and the reasons for choosing it should also be described in detail.
8) The description of the formula ( Line 136: Dietary intake ng/Kg/day) in the text (lines 132-134) is unclear.
9) The statistical approach should be improved and better described. Methods and indicators of statistical analysis should be described in detail, taking into account the design of the study.
10) Line 146. “The coefficient of determination was R2 0.9985”. Please specify for which indicator and by what method R2 was calculated.
11) Line 156. Title "3.2. Milk and its products" does not reflect the content.
12) Lines Table 3 . % sample must be specified everywhere the same - one decimal place. The same remark applies to table 4.
13) Data “2.5. Dietary Intake” described in the methodology were not found in the results section.
14) I recommend that the authors thoroughly review the objectives of the study and adjust the design, study results, discussion, and conclusions accordingly.
15) In the bibliography, the unequal presentation of journal titles should be noted. Some titles are in italics, others are not.
16) The article is interesting, but it is not clear how it can be relevant for researchers and practitioners in other countries.
Sincerely, reviewer.
Round 2
Reviewer 1 Report
Dear authors,
the paper was improved after all the reviewers comments, now it is clearly written and in good condition.
Author Response
The attached file

Reviewer 4 Report
I thank the authors for taking into account all the comments and significantly improving the quality of the article.
Author Response
The attached file
